# Patient and health system related factors affecting diagnostic and treatment delays in tuberculosis: A cross-sectional study in Mysuru, South India

Veerabhadra Swamy G S[1], Mahesh Padukudru Anand[2]*,
Mahadevaiah Neelambike Sumana[1]*, N B Ramachandra[3], Prashanth Chikkahonnaiah[4],
Ranjitha Shankaregowda[1], Jyothi M N[5], Chinchana Shylaja Eshwarappa[1],
Yogeesh D Maheshwarappa[1], Jayaraj B S[2]

1 Department of Microbiology, JSS Medical College and Hospital, JSS Academy of Higher Education and Research, Mysuru, India, 2 Department of Pulmonary Medicine, JSS Medical College and Hospital, JSS Academy of Higher Education and Research, Mysuru, India, 3 Department of Studies in Genetics & Genomics, University of Mysore, Mysuru, India, 4 Department of Respiratory Diseases, Princess Krishnajammanni Tuberculosis and Chest Diseases Hospital, Mysuru, India, 5 Department of Medical Genetics, JSS Medical College and Hospital, JSS Academy of Higher Education and Research, Mysuru, India

* mahesh1971in@yahoo.com (PA); mnsumana@jssuni.edu.in (MN)

## Abstract

Tuberculosis (TB) remains a major public health challenge in India despite extensive efforts under the National TB Elimination Programme. Delays in diagnosis and treatment initiation contribute substantially to ongoing transmission and poor clinical outcomes. This study assessed the duration and determinants of patient, health system, and total delays among TB patients in Mysuru, South India. A hospital based cross-sectional retrospective study was conducted at the Princess Krishnajammanni Tuberculosis and Chest Diseases (PKTB) Sanatorium, Mysuru, between January and June 2025. A total of 331 microbiologically confirmed TB patients were interviewed using a pretested semi-structured questionnaire. Delays were defined as patient delay (>30 days from symptom onset to first healthcare consultation), health system delay (>7 days from the first consultation to treatment initiation), and total delay (>37 days). Logistic regression analysis was used to identify factors associated with prolonged delays. The median (IQR) durations for patient, health system, and total delays were 30 (25), 12 (16), and 43 (32) days, respectively. Patient delay was independently associated with informal education (AOR: 1.95; 95% CI: 1.02–3.71) and poor TB-related knowledge (AOR: 31.40; 95% CI: 3.46–284.74). Health system delay was associated with socioeconomic vulnerability, including daily wage occupation (AOR: 6.15; 95% CI: 1.81–20.85), below-poverty-line economic status (AOR: 13.89; 95% CI: 3.38–57.17), first seeking care at private facilities (AOR: 17.28; 95% CI: 5.86–50.94), and visiting more than three healthcare facilities before diagnosis

**Data availability statement:** The data underlying the results presented in the study are available from Zenodo (https://doi.org/10.5281/zenodo.18477319).

**Funding:** The author(s) received no specific funding for this work.

**Competing interests:** The authors have declared that no competing interests exist.

(AOR: 3.01; 95% CI: 1.23–7.32). Prolonged total delay (>37 days) was associated with informal education (AOR: 4.91; 95% CI: 1.64–14.76), being married (AOR: 4.34; 95% CI: 1.12–16.85), consulting private facilities initially (AOR: 15.80; 95% CI: 5.45–45.78), and multiple healthcare visits prior to diagnosis (AOR: 24.81; 95% CI: 10.34–59.50). Nearly two-thirds of TB patients experienced diagnostic delay. Poor awareness, socioeconomic disadvantage, and initial consultations at private facilities were the major contributors. Strengthening community awareness, improving engagement with private practitioners, and addressing socioeconomic barriers are essential to reducing delays and supporting India's TB elimination goals.

## Introduction

Tuberculosis (TB) is a chronic infectious disease caused by *Mycobacterium tuberculosis*, primarily affecting the lungs but capable of spreading to other organs through the bloodstream if untreated. Transmission occurs via aerosolized droplets expelled during coughing, sneezing, or speaking by individuals with infectious pulmonary TB. Despite sustained global control efforts, TB remains a major public health concern, particularly in low- and middle-income countries (LMICs) where early diagnosis and treatment initiation continue to be challenging [1,2].

According to the World Health Organization (WHO) Global Tuberculosis Report 2024 [3], about 10.8 million people developed TB in 2023, resulting in an estimated 1.25 million deaths. India accounts for 26% of the global TB burden and 25.86% of TB-related deaths, making it the highest-burden country. Although a 17.7 percent reduction in incidence has been achieved since 2015, approximately 2.7 million new cases were reported in 2023 [3,4]. TB remains the second leading infectious cause of death globally after COVID-19, surpassing HIV/AIDS [5].

India's National TB Elimination Programme (NTEP), formerly known as the Revised National TB Control Programme (RNTCP), has made significant progress through the Directly Observed Treatment Short-course (DOTS) strategy. However, diagnostic delays remain a major obstacle to achieving elimination goals [6,7]. The WHO has projected that India is unlikely to meet its TB elimination target by 2025, falling short of milestones that aimed for a 75% reduction in deaths and a 50% reduction in incidence compared with 2015 levels [3].

The WHO End TB Strategy emphasizes early diagnosis and prompt treatment to reduce transmission and mortality, targeting a 95% reduction in deaths and a 90% reduction in incidence by 2035, along with the elimination of catastrophic costs for affected households [8,9]. In alignment with this framework, India has set national targets to achieve an 80% reduction in incidence and a 90% reduction in mortality by 2030 compared with 2015 levels [10].

Despite these initiatives, diagnostic and treatment delays continue to be reported in high-burden countries such as India [2,11–19]. Patient delay is often associated with stigma, low awareness, limited income, and reliance on informal healthcare providers. Health system delay results from missed opportunities for early diagnosis and

dependence on passive case detection. These delays prolong infectivity, worsen disease severity, and sustain community transmission. An untreated case can result in up to 28 secondary infections [5]. Studies from India estimate a median total delay of approximately 55 days from symptom onset to treatment initiation, including a patient delay of around 18 days and a diagnostic delay of about 31 days [14]. Such prolonged delays extend the infectious period, increase the risk of transmission, and contribute to poor treatment outcomes and drug resistance.

The present study aims to assess the duration from the onset of TB symptoms to confirmed diagnosis and to identify factors independently associated with diagnostic and treatment delays among TB patients. The findings are expected to provide evidence for strengthening early case detection, improving patient outcomes, and guiding more effective TB control strategies in high-burden settings.

## Materials and methods

### Study design and setting

This hospital-based cross-sectional retrospective study was conducted over six months (January to June 2025) at the Princess Krishnajammanni Tuberculosis and Chest Diseases (PKTB) Sanatorium in Mysore, Karnataka, India [20]. PKTB functions as a specialized secondary-care center for tuberculosis and other chest diseases, which operates under the District Tuberculosis Centre (DTC) and is affiliated with the National Tuberculosis Elimination Programme (NTEP), providing both outpatient (OPD) and inpatient (IPD) services to Mysore and surrounding districts [21–23]. This study analyzed microbiologically confirmed pulmonary and extrapulmonary TB cases diagnosed and managed according to NTEP guidelines during the study period using the sanatorium's facilities. The retrospective design enabled assessment of TB burden, treatment and patient demographics within a public health context, contributing valuable insights into TB control efforts in Mysore district.

### Study participants

Participants were enrolled consecutively from both OPD and IPD services at PKTB Hospital. The study population comprised newly diagnosed TB patients aged ≥18 years who gave written informed consent and were available for interviews during the study period. Patients were excluded if they declined consent, could not be reached after two follow-up attempts, had a history of recurrent TB infection, or were undergoing TB treatment at the time of enrolment.

### Sample size and selection

The sample size for the study was calculated using the single population proportion formula [24], with a prevalence of approximately 40.5% for delays of more than one month, based on combined data from studies conducted in Mysore, Karnataka, and other regions of India [12,25–27]. Using a 95% confidence interval (Z = 1.96) and a 5% absolute precision, the initial sample size was estimated to be 370 participants. Given a finite eligible population of 325, the sample size was adjusted to 173 using the finite population correction formula. Accounting for a 10% nonresponse rate, the final calculated sample size was approximately 190 participants. However, to enhance statistical power and ensure representativeness across the targeted population, the study included 331 participants. The flow of participant selection and inclusion in the study is presented in Fig 1.

### Data collection techniques and tools

Data were collected through face-to-face interviews conducted at the time of diagnosis or shortly thereafter, in either Kannada or English, according to participant preference. A pretested semi-structured questionnaire (Supplementary File-1) was employed, which had been developed through a literature review and adaptation of TB stigma and patient knowledge items from the World Health Organization's multi-country study on diagnosis and treatment delays in tuberculosis,

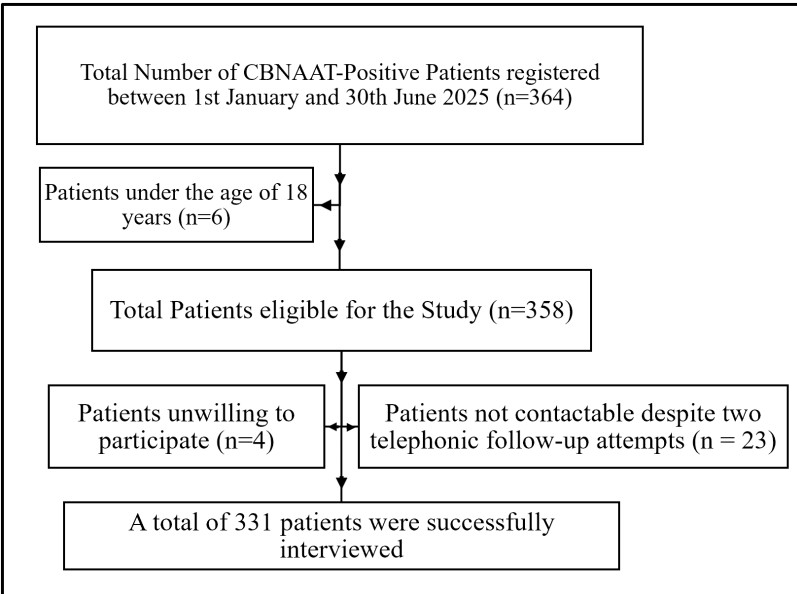

**Fig 1. Flow of participants.**

as well as tools from studies on TB diagnostic delay in India and other high-burden settings [2,13,14,16,28]. Patient recall of symptom onset and care-seeking was aided by memory cues such as significant events or calendar references, and details were cross-verified with data available on NTEP/NIKSHAY portal records and patients' health documents. Provider-related factors, including low clinical suspicion, misdiagnosis, unnecessary antibiotic use before TB testing, and delayed referrals, were assessed via patient reports and verified against health records such as prescriptions and referral slips. All terms were pre-defined for clarity and consistency. The questionnaire underwent review, verification, and back-translation (English–Kannada–English) to ensure clarity, cultural appropriateness, and content validity. Each interview lasted approximately 20–25 minutes.

### Study variables

**Independent variables.** Socio demographic, economic variables, Health-seeking behavior was evaluated by asking patients about their initial response following symptom onset and timelines, types of treatment sought at the first time, the type of facility they first visited, number and type of providers and barriers to early diagnosis and treatment. Clinical characteristics included the type of TB (pulmonary or extrapulmonary), presence of HIV or other chronic comorbidities, the type of health facility where the final diagnosis was established, and smear status (positive or negative).

Variables such as occupation, marital status, residence, and economic status were treated as patient-level determinants that may influence delays occurring within the health system after initial care-seeking, rather than as intrinsic components of the health system itself. These factors were included in the health system delay analysis to assess how socioeconomic vulnerability affects patients' ability to navigate diagnostic pathways, attend repeated visits, and access timely investigations and treatment within the healthcare system. TB-related knowledge: Participants' knowledge of TB was assessed using 15 questions addressing knowledge about TB, contagiousness, curability, vaccine availability, and treatment duration [28,30]. Correct answers were scored as one, while incorrect or "do not know" responses were scored as zero. Item scores were summed to generate a total knowledge score ranging from 0 to 15. The median score of 7 was used as the cut-off, with respondents scoring ≥7 categorized as having good knowledge and those scoring <7 as having poor knowledge, the scoring approach was adapted from a previous study [31].

Perceived stigma: Stigma associated with TB among the study participants was assessed using 15 negative statements covering aspects such as shame, concealment of diagnosis, financial burden from prolonged illness, social isolation, girls' autonomy in treatment decisions, and the perceived impact of tuberculosis on interpersonal relations, work performance, marital life, family responsibilities, marriage prospects, family relations, female infertility, pregnancy complications, breastfeeding, and pregnancy outcomes [28,30]. Each response indicating absence of stigma was scored as one, and presence of stigma as zero. Scores ranged from 0 to 15, with higher values reflecting lower stigma. The overall score was dichotomized at the median (5), classifying respondents with scores ≥5 as having low stigma and those with scores <5 as having high stigma, following methods used in a previous study [31].

Smoking status was categorized based on WHO and CDC criteria as current smokers (those who smoked any tobacco product daily or occasionally at the time of interview), former smokers (those who had quit smoking for at least six months), and never smokers (those who had never smoked in their lifetime) [32,33]. Alcohol consumption was defined according to WHO guidelines as current users (those who had consumed alcohol within the past 12 months), past users (those who had abstained for at least 12 months), and never users (those who had never consumed alcohol) [34]. Body mass index (BMI) was calculated as weight in kilograms divided by height in meters squared (kg/m²) and classified as underweight (<18.5 kg/m²), normal (18.5–24.9 kg/m²), overweight (25.0–29.9 kg/m²), or obese (≥30.0 kg/m²) according to WHO standards [35].

**Dependent variables** are Patient delay, health system delay and Total delay.

**Operational definitions:**

- **Patient delay**: The time interval (in days) between the onset of symptoms and first presentation to a healthcare provider, with >30 days considered unacceptable [12,28].

- **Health system delay**: The time interval (in days) between the patient's first health-seeking encounter and initiation of treatment, with >7 days considered unacceptable [12,28].

- **Diagnostic delay**: The time interval (in days) between the onset of TB-related symptoms and confirmation of diagnosis, which may be due to patient delay or health system delay [28].

- **Treatment delay:** The time interval (in days) between the confirmation of diagnosis and initiation of treatment, which reflects only health system delay. This was considered unacceptable if more than seven days [14,29].

- **Total delay**: The sum of patient delay and health system delay, was deemed unacceptable if >37 days [12,36], as depicted in Figure 2.

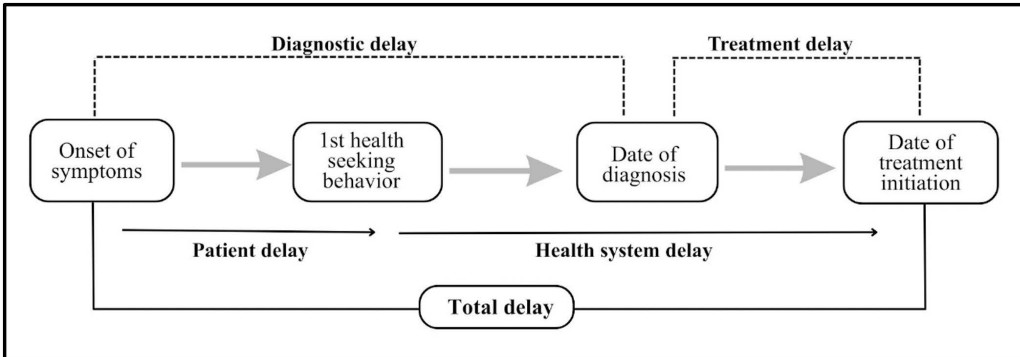

**Fig 2. Delays in diagnosis and treatment among tuberculosis patients.**

## Data quality assurance

Pretesting was conducted among 25 TB patients recruited from JSS Hospital and Krishna Rajendra Hospital in Mysuru, and these data were not included in the final analysis. Following the pretest, the sequence of questions was rearranged, and items that participants found difficult to understand were rephrased. During data collection, questionnaires were checked immediately after each interview for completeness and consistency, with discrepancies subsequently clarified through follow-up calls. Patients' health records were also cross-checked to ensure data quality. This three-step validation process enhanced the reliability of the interview responses in reflecting participant history and outcomes. Internal consistency of the instrument, assessed using Cronbach's alpha, was 0.73 for knowledge and 0.85 for stigma. In addition, the questionnaire content was reviewed by subject experts to ensure face and content validity.

## Data management and statistical analysis

The collected data were entered into Microsoft Excel and double-checked to minimize human error during entry. The verified dataset was then exported to IBM SPSS Statistics Version 20.0 for further analysis. Categorical variables were summarized as frequencies and percentages, while continuous variables were described using the mean, standard deviation, median, and interquartile range (minimum–maximum). Delays in tuberculosis (TB) diagnosis were dichotomized using 30 days for patient delay and 7 days for health system delay, based on clinical relevance and previous studies from India and Nepal [12,28]. Associations between the dependent variable and independent variables were quantified using odds ratios (ORs) with 95% confidence intervals. Independent variables with $p \leq 0.05$ in the bivariate analysis were included in the multivariable logistic regression model. The significance of variables in the multivariable model was assessed using adjusted odds ratios (AORs) and 95% confidence intervals. Model fitness was assessed using the Hosmer-Lemeshow chi-square goodness-of-fit test, with p-values greater than 0.05 indicating an adequate fit. A two-tailed p-value of 0.05 or less was considered statistically significant.

## Ethical considerations

Ethical approval for the study was obtained from thse Institutional Ethics Committee, JSS Medical College and Hospital, Mysuru (JSSMC/IEC/07112024/09 NCT/2024–25). Permission to conduct the study at the PKTB Sanatorium was obtained from the District Tuberculosis Officer (DTO), Mysore and also from the State Tuberculosis Officer (STO). All participants were informed about the purpose of the study, and written informed consent was obtained prior to data collection. Interviews were conducted in a private area within the health facility to maintain confidentiality. Participants were assigned numerical codes in place of names to ensure anonymity. Participation was voluntary, with participants informed that they could skip questions or withdraw from the interview at any time.

# Results

## Sociodemographic characteristics

The sociodemographic and lifestyle characteristics of the participants are presented in Table 1. Overall, the study population was predominantly male, aged between 20 and 60 years, and largely from rural areas. Most participants belonged to the below-poverty-line category, with daily wage employment being common. Tobacco and alcohol use were frequently reported.

## Clinical characteristics

Cough was the most frequently reported symptom (88.22%), of which 63.36% produced non-bloody sputum, followed by weight loss (64.04%), fatigue (51.06%), and loss of appetite (47.43%). Night sweats (36.25%) and fever (34.74%) were also common, with nearly half (46.09%) of febrile patients reporting an evening rise of temperature. Most participants

**Table 1. Sociodemographic and lifestyle-related characteristics of tuberculosis patients (n = 331).**

| Variables | Frequency (n) | Percentage (%) |
|---|---|---|
| **Age group** | | |
| <20 Yrs. | 6 | 1.81 |
| 20-60 Yrs. | 258 | 77.95 |
| >60 Yrs. | 67 | 20.24 |
| **Sex** | | |
| Male | 253 | 76.44 |
| Female | 78 | 23.56 |
| **Religion** | | |
| Hindu | 290 | 87.61 |
| Muslim | 40 | 12.08 |
| Christian | 1 | 0.30 |
| **Residence** | | |
| Rural | 210 | 63.44 |
| Urban | 121 | 36.56 |
| **Income status** | | |
| BPL | 293 | 88.52 |
| APL | 26 | 7.85 |
| Not available | 12 | 3.63 |
| **Marital status** | | |
| Married | 286 | 86.40 |
| Unmarried | 45 | 13.60 |
| **Education** | | |
| Illiterate | 126 | 38.07 |
| Primary (1 to 5th standard) | 67 | 20.24 |
| Middle school (6th to 8th standard) | 60 | 18.13 |
| Secondary (9 to 10th standard) | 35 | 10.57 |
| Senior secondary (11 and 12th or PUC) | 28 | 8.46 |
| Graduate | 15 | 4.53 |
| **Occupation** | | |
| Unemployed | 17 | 5.14 |
| Agriculture | 57 | 17.22 |
| Salaried | 34 | 10.27 |
| Business | 43 | 12.99 |
| Daily wage/casual | 71 | 21.45 |
| Housewife | 53 | 16.01 |
| Dependent | 45 | 13.60 |
| Students | 11 | 3.32 |
| **Health Insurance Enrolment Status** | | |
| yes | 26 | 7.85 |
| No | 305 | 92.15 |
| **Type of Family** | | |
| Nuclear | 277 | 83.69 |
| Joint/Extended | 54 | 16.31 |
| **Family size (Members)** | | |
| 1 to 4 | 170 | 51.36 |
| 5 to 7 | 125 | 37.76 |

*(Continued)*

Table 1. (Continued)

| Variables | Frequency (n) | Percentage (%) |
|---|---|---|
| >8 | 36 | 10.88 |
| **Family H/O TB** | | |
| Yes | 17 | 5.14 |
| No | 314 | 94.86 |
| **Previous contact with TB patient** | | |
| Yes | 32 | 9.67 |
| No | 179 | 54.08 |
| Not known | 120 | 36.25 |
| **Smoker** | | |
| Current smoker | 30 | 9.06 |
| Quitted smoking | 141 | 42.60 |
| Non-smokers (never) | 160 | 48.34 |
| **Alcoholic** | | |
| Current user | 54 | 16.31 |
| Past user | 152 | 45.92 |
| Never | 125 | 37.76 |
| **BMI** | | |
| Normal | 118 | 35.65 |
| Underweight | 189 | 57.10 |
| Overweight | 22 | 6.65 |
| Obese | 2 | 0.60 |

(95.77%) had pulmonary tuberculosis, while 4.23% presented with extra-pulmonary disease. Sputum smear positivity was observed in nearly three-fourths (71.00%) of cases. CBNAAT results showed a very low bacterial load in more than one-third (37.16%) and a high load in about one-fourth (23.26%). The majority (92.10%) tested negative for HIV, and 6.60% had unknown status. Slightly more than one-fourth (28.08%) reported comorbidities other than HIV (Table 2).

Among the 14 extrapulmonary tuberculosis cases, pleural TB was most common (n = 8), followed by lymph node TB (n = 4) and abdominal TB (n = 2). Night sweats were reported in all patients, and fever in 71.4%, mostly intermittent. Weight loss and fatigue were observed in 57.1% and 50%, respectively. Respiratory symptoms were uncommon; cough occurred in four patients (mainly dry), and chest pain and breathlessness only in pleural TB. Lymph node TB presented with pain-less lymphadenopathy, while abdominal TB presented with chronic abdominal pain; both were associated with constitutional symptoms but lacked respiratory features.

## Health-seeking behavior

Nearly half of the participants (45.62%) reported visiting a healthcare facility as their first action to illness, followed by self-medication (26.28%), pharmacy consultation (17.52%), and use of traditional medicine (6.95%). Among medical facilities, health centers were the most frequently chosen, with more than half (52.87%) initially visiting them. Most participants (88.82%) had visited multiple healthcare facilities before receiving a final diagnosis, and over half (55.89%) had made more than three visits. The majority (70.09%) reached a healthcare facility within 30 minutes. About one-third (32.33%) received a diagnosis within 7 days of their first visit, and only 29.31% initiated treatment within this period, while 70.69% experienced a delay. Following a confirmed diagnosis, nearly all participants (93.96%) began treatment within 7 days, with only a few (6.04%) starting later (Table 3).

**Table 2. Clinical characteristics of tuberculosis patients (n = 331).**

| Variables | Frequency (n = 331) | Percentage (%) |
|---|---|---|
| **Clinical sign and symptoms** | | |
| Cough | 292 | 88.22 |
| Dry cough (n = 292) | 88 | 30.14 |
| Cough productive of bloody sputum (n = 292) | 19 | 6.51 |
| Cough productive of non-bloody sputum (n = 292) | 185 | 63.36 |
| Weight loss | 212 | 64.04 |
| Fatigue | 169 | 51.06 |
| Loss of appetite | 157 | 47.43 |
| Night sweats | 120 | 36.25 |
| Fever | 115 | 34.74 |
| Persistent fever (n = 115) | 12 | 10.43 |
| Evening rise of temperature (n = 115) | 53 | 46.09 |
| Intermittent fever (n = 115) | 36 | 31.30 |
| Intermittent and evening rise of temperature (n = 115) | 14 | 12.17 |
| Chest pain | 76 | 22.96 |
| Breathlessness | 57 | 17.22 |
| **Type of TB disease** | | |
| Pulmonary | 317 | 95.77 |
| Extra-pulmonary | 14 | 4.23 |
| **HIV status** | | |
| Reactive | 4 | 1.20 |
| Non-Reactive | 305 | 92.10 |
| Unknown | 22 | 6.60 |
| **Other comorbidities** | | |
| Diabetes | 54 | 16.30 |
| Hypertension | 39 | 11.80 |
| **Sputum examination result** | | |
| Positive | 235 | 71.00 |
| Negative | 42 | 12.70 |
| Not done | 54 | 16.30 |
| **Smear grading (n = 235)** | | |
| Negative | 42 | 17.87 |
| Scanty | 46 | 19.57 |
| 1+ | 50 | 21.27 |
| 2+ | 52 | 22.12 |
| 3+ | 87 | 37.02 |
| **Bacterial load (CBNAAT)** | | |
| High | 77 | 23.26 |
| Medium | 63 | 19.03 |
| Very low | 123 | 37.16 |
| Low | 68 | 20.54 |

**Table 3. Health seeking behavior among Tuberculosis patients (n = 331).**

| Variable | | Frequency (n) | Percentage (%) |
|---|---|---|---|
| First action to illness | Visit Health-Care Facility | 151 | 45.62 |
| | Self-medication | 87 | 26.28 |
| | Consult Health worker | 12 | 3.63 |
| | Pharmacy | 58 | 17.52 |
| | Use traditional medicine | 23 | 6.95 |
| Facility first visited | Private facilities | 41 | 12.39 |
| | Health centre (Clinic) | 175 | 52.87 |
| | Public hospital | 78 | 23.56 |
| | Tuberculosis Treatment Centre | 37 | 11.18 |
| Time spent from the onset of symptoms to the first HCF visit (days) | ≤30 | 232 | 70.09 |
| | >30 | 99 | 29.91 |
| No. of HCF Visited | Single | 37 | 11.18 |
| | multiple | 294 | 88.82 |
| Number of visits to HCF | ≤3 | 146 | 44.11 |
| | >3 | 185 | 55.89 |
| Place of TB diagnosis | Private | 29 | 8.76 |
| | Public | 302 | 91.24 |
| Distance of nearest public health facility from home (km) | <3 km | 135 | 40.79 |
| | >3 km | 196 | 59.21 |
| Mode of transport | Walk | 15 | 4.53 |
| | Private vehicle | 145 | 43.81 |
| | Public transport | 171 | 51.66 |
| Time spent from first HCF visit to Final Diagnosis (days) | ≤7 | 107 | 32.33 |
| | >7 | 224 | 67.67 |
| Time spent initiating treatment since the first HCF visit (days) | ≤7 | 97 | 29.31 |
| | >7 | 234 | 70.69 |
| Time spent for initiation of Treatment after Final Diagnosis (days) | ≤7 | 311 | 93.96 |
| | >7 | 20 | 6.04 |

## Reasons for diagnostic and treatment delay, Knowledge, and stigma in tuberculosis patients

The most common patient-related factors for diagnostic delay were low awareness of TB symptoms (68.88%) and stigma or fear of isolation (62.54%) (Tables 4 and 5). Economic barriers such as poverty or low income (33.32%) and catastrophic costs (29.00%) were also reported. Other factors included busy occupations (12.99%), transportation or long distance (7.85%), and cultural beliefs (6.95%). For treatment delay, poor knowledge of TB treatment protocols (6.95%) was most common, with smaller proportions reporting long distance to health facilities (3.02%) or reluctance and fear of prolonged treatment (<2%) (Table 5).

Among provider-related factors, low clinical suspicion of tuberculosis was most frequent (58.00%), followed by misdiagnosis (12.99%) and unnecessary antibiotic use before TB testing (12.39%), while delayed referrals were uncommon (1.51%). Treatment delays related to providers included loss to follow-up after diagnosis (6.65%) and logistical or administrative delays (1.51%) (Table 5).

## Delays in diagnosis and treatment among tuberculosis patients

The average time for patients to first visit a healthcare provider after symptom onset was 31.05 ± 14.74 days (median: 30; IQR: 25). The health system added a mean delay of 14.93 ± 12.77 days (median: 12; IQR: 16) from first contact to

**Table 4. Knowledge and perceived stigma related to tuberculosis patients (n = 331).**

| Variable | Frequency (n) | Percentage (%) |
|---|---|---|
| **TB-related Knowledge** | | |
| Low/Poor | 228 | 68.88 |
| Good | 103 | 31.12 |
| **Perceived Stigma** | | |
| Low | 124 | 37.46 |
| High | 207 | 62.54 |

**Table 5. Determinants of Diagnostic and Treatment Delays in Tuberculosis Patients (n = 331).**

| Reasons | | Frequency (n) | Percentage (%) |
|---|---|---|---|
| **Diagnostic delay** | | | |
| **Patient-related factors** | | | |
| Economic factors | Poverty and low income | 11 | 3.32 |
| | Unemployment | 22 | 6.65 |
| | Catastrophic costs | 96 | 29.00 |
| Social factors | Low awareness of TB symptoms | 228 | 68.88 |
| | Stigma and fear of isolation | 207 | 62.54 |
| | Crowded living conditions | 19 | 5.74 |
| | Gender disparities | 17 | 5.14 |
| | Cultural beliefs and practices | 23 | 6.95 |
| | Busy occupation | 43 | 12.99 |
| | Transportation and long distance | 26 | 7.85 |
| Others | Delay in seeking care from a higher-level healthcare provider after leaving a previous provider | 22 | 6.65 |
| | migration | 8 | 2.42 |
| **Provider-related factors** | | | |
| Low clinical suspicion of tuberculosis | | 192 | 58.00 |
| Prescribing unnecessary antibiotics before TB tests | | 41 | 12.39 |
| Misdiagnosis | | 43 | 12.99 |
| Delayed referrals | | 5 | 1.51 |
| **Treatment delay** | | | |
| **Patient-related factors** | | | |
| Poor knowledge of TB treatment protocols | | 23 | 6.95 |
| Long distance from health facility | | 10 | 3.02 |
| Reluctance to initiate TB treatment due to its prolonged duration | | 5 | 1.51 |
| Fear of long treatment | | 2 | 0.60 |
| **Provider-related factors** | | | |
| Delayed initiation after diagnosis due to administrative or logistical issues | | 5 | 1.51 |
| Loss to follow-up after TB diagnosis | | 22 | 6.65 |

treatment initiation. Overall, the total delay from symptom onset to treatment initiation was 47.85 ± 22.53 days (median: 43; IQR: 32) (Table 6).

## Factors associated with patient delay

In the multivariable logistic regression analysis, informal education (AOR: 1.95; 95% CI: 1.02–3.71) and poor knowledge of TB symptoms (AOR: 31.40; 95% CI: 3.46–284.74) were independently associated with higher odds of patient delay

**Table 6. Distribution of various delays (days) among TB patients (n = 331).**

| Type of delay | Mean;95% CI (SD) | Median (IQR) | Range (minimum-maximum) |
|---|---|---|---|
| Patient delay | 31.05; 29.46–32.65 (14.74) | 30 (25) | 83 (7–90) |
| Health system delay | 14.93; 13.55–16.31 (12.77) | 12 (16) | 66 (1–67) |
| Diagnostic delay | 45.98; 43.66–48.30 (21.47) | 41 (30) | 94 (14–108) |
| Treatment delay | 1.87; 1.25–2.49 (5.70) | 1 (1) | 66 (0–66) |
| Total delay | 47.85; 45.42–50.29 (22.53) | 43 (32) | 111 (14–125) |

beyond 30 days (Table 7). These large AORs and wide confidence intervals should be interpreted with caution due to sparse data in certain categories.

### Factors associated with health system delay

Multivariable logistic regression showed that health system delay beyond 7 days was independently associated with being a daily wage or casual worker (AOR: 6.15; 95% CI: 1.81–20.85), belonging to the below poverty line category (AOR: 13.89; 95% CI: 3.38–57.17), first visiting a private healthcare facility (AOR: 17.28; 95% CI: 5.86–50.94), visiting healthcare facilities more than three times before TB diagnosis (AOR: 3.01; 95% CI: 1.23–7.32), and being diagnosed at a private facility (AOR: 0.072; 95% CI: 0.018–0.30) (Table 8).

### Factors associated with total delay among tuberculosis patients

Seeking care first at private healthcare facilities (AOR: 15.80; 95% CI: 5.45–45.78) and visiting healthcare facilities more than three times before diagnosis (AOR: 24.81; 95% CI: 10.34–59.50) were significantly associated with total delays exceeding 37 days. Being married (AOR: 4.34; 95% CI: 1.12–16.85) and having informal education (AOR: 4.91; 95% CI: 1.64–14.76) also significantly increased the odds of prolonged delay (Table 9).

## Discussion

Timely diagnosis and prompt treatment are essential for effective TB control. Understanding the factors driving patient and health system delays is critical for identifying barriers and informing strategies to improve the quality and efficiency of TB care. This study examined delays in TB diagnosis and treatment and their associated factors among patients in Mysuru, South India. The median (IQR) patient and health system delays were 30 (25) and 12 (16) days, respectively. Patient delays beyond the median were significantly associated with informal education and poor knowledge of TB symptoms. Factors contributing to prolonged health system delays included being a daily wage or casual worker, belonging to the below-poverty-line category, first seeking care at private healthcare facilities, visiting healthcare facilities more than three times before diagnosis, and receiving a TB diagnosis at a private facility. Unacceptable total delays were significantly linked to initially seeking care at private healthcare facilities, consulting multiple healthcare facilities before diagnosis, being married, and having informal education. Although large AORs were observed, the wide confidence intervals indicate imprecision due to sparse data, and therefore these estimates should be interpreted cautiously.

### Factors associated with patient delay

This study demonstrated a considerable delay in TB diagnosis in Mysuru Province, with a median diagnostic delay of 30 days, exceeding the recommended 14–21 days from symptom onset [37]. Similar findings have been reported in Karnataka, India, and other countries [11–13,16,25,26,28,36,38–40], where the median patient delay was approximately 30 (±5) days. Sreeramareddy et al. also reported a mean delay of 31.7 days in LMICs [41]. In contrast, several studies

**Table 7. Factors associated with patient delay among tuberculosis patients (n = 331).**

| Variables | ≤30 days (n = 232) | >30 days (n = 99) | Total (n = 331) | Unadjusted OR (95% CI) | p-value | AOR (95% CI) | p-value |
|---|---|---|---|---|---|---|---|
| **Age group** | | | | | | | |
| 19–60 yrs | 193 (73.38) | 70 (26.62) | 263 (79.46) | **Ref.** | **0.01** | Ref. | 0.304 |
| >60 yrs | 39 (57.35) | 29 (42.65) | 68 (20.54) | **2.05 (1.17–3.56)** | | 0.69 (0.34-1.39) | |
| **Sex** | | | | | | | |
| Male | 175 (69.17) | 78 (30.83) | 253 (76.44) | Ref. | 0.51 | | |
| Female | 57 (73.08) | 21 (26.92) | 78 (23.56) | 0.82 (0.46–1.45) | | | |
| **Type of Residence** | | | | | | | |
| Rural | 144 (68.57) | 66 (31.43) | 210 (63.44) | 1.22 (0.74–2.00) | 0.426 | | |
| Urban | 88 (72.73) | 33 (27.27) | 121 (36.56) | Ref. | | | |
| **Marital status** | | | | | | | |
| Married | 189 (66.08) | 97 (33.92) | 286 (86.4) | **11.03 (2.61–46.51)** | **<0.001** | 1.60 (0.29-8.67) | 0.582 |
| Unmarried | 43 (95.56) | 2 (4.44) | 45 (13.6) | **Ref.** | | Ref. | |
| **Education** | | | | | | | |
| Formal | 166 (80.98) | 39 (19.02) | 205 (61.93) | **Ref.** | **<0.001** | **Ref.** | **0.043** |
| Informal | 66 (52.38) | 60 (47.62) | 126 (38.07) | **3.86 (2.36-6.34)** | | **1.94 (1.02-3.71)** | |
| **Occupation** | | | | | | | |
| Daily wage/casual | 40 (56.34) | 31 (43.66) | 71 (21.45) | **2.18 (1.26-3.77)** | **0.005** | 1.05 (0.56-1.97) | 0.861 |
| Others | 192 (73.85) | 68 (26.15) | 260 (78.55) | **Ref.** | | Ref. | |
| **Income status (n = 319)** | | | | | | | |
| Below poverty line (BPL) | 196 (66.89) | 97 (33.11) | 293 (88.52) | **12.37 (1.65–92.66)** | **0.014** | 0.34 (0.02-5.77) | 0.458 |
| Above poverty line (APL) | 25 (96.15) | 1 (3.85) | 26 (7.85) | **Ref.** | | Ref. | |
| **Health Insurance Enrolment Status** | | | | | | | |
| Yes | 15 (57.69) | 11 (42.31) | 26 (7.85) | Ref. | 0.153 | | |
| No | 217 (71.15) | 88 (28.85) | 305 (92.15) | 0.55 (0.24–1.25) | | | |
| **Type of Family** | | | | | | | |
| Nuclear | 196 (70.76) | 81 (29.24) | 277 (83.69) | Ref. | 0.548 | | |
| Joint/Extended | 36 (66.67) | 18 (33.33) | 54 (16.31) | 1.21 (0.64–2.25) | | | |
| **Facility first visited** | | | | | | | |
| Private | 154 (71.30) | 62 (28.70) | 216 (65.26) | 0.84 (0.52-1.38) | 0.512 | | |
| Public | 78 (67.83) | 37 (32.17) | 115 (34.74) | Ref. | | | |
| **Distance of nearest public health facility from home (km)** | | | | | | | |
| ≤3 | 96 (71.11) | 39 (28.89) | 135 (40.79) | Ref. | 0.736 | | |
| >3 | 136 (69.39) | 60 (30.61) | 196 (59.21) | 1.08 (0.67 −1.75) | | | |
| **Knowledge score towards TB symptoms** | | | | | | | |
| Poor | 132 (57.89) | 96 (42.11) | 228 (68.88) | **24.24 (7.46–78.74)** | **<0.001** | **31.40 (3.46-284.74)** | **0.002** |
| Good | 100 (97.09) | 3 (2.91) | 103 (31.12) | **Ref.** | | **Ref.** | |
| **Stigma Score** | | | | | | | |
| Low | 114 (91.94) | 10 (8.06) | 124 (37.46) | **Ref.** | **<0.001** | Ref. | 0.966 |
| High | 118 (57.00) | 89 (43.00) | 207 (62.54) | **8.59 (4.25–17.35)** | | 1.02 (0.35-2.94) | |
| **Smoking status** | | | | | | | |
| Yes | 108 (63.16) | 63 (36.84) | 171 (51.66) | **2.00 (1.23–3.26)** | **0.005** | 1.15 (0.66-2.01) | 0.616 |
| No | 124 (77.50) | 36 (22.50) | 160 (48.34) | **Ref.** | | **Ref.** | |
| **Alcoholic status** | | | | | | | |
| Yes | 138 (66.99) | 68 (33.01) | 206 (62.24) | 1.49 (0.90–2.46) | 0.115 | | |
| No | 94 (75.20) | 31 (24.80) | 125 (37.76) | Ref. | | | |

\* Only variables significant in univariate analysis were included in the multivariable model; blanks indicate variables not included.

**Table 8. Factors associated with health system delay among tuberculosis patients (n = 331).**

| Variables | ≤7 (n = 97) | >7 (n = 234) | Total (n = 331) | Unadjusted OR (95% CI) | p-value | AOR (95% CI) | p-value |
|---|---|---|---|---|---|---|---|
| **Age group** | | | | | | | |
| 19-60 Yrs. | 69 (26.24) | 194 (73.76) | 263 (79.46) | **Ref** | **0.017** | Ref | 0.341 |
| >60 Yrs. | 28 (41.18) | 40 (58.82) | 68 (20.54) | **0.50 (0.29–0.88)** | | 1.64 (0.58-4.62) | |
| **Sex** | | | | | | | |
| Male | 79 (31.23) | 174 (68.77) | 253 (76.44) | Ref | 0.169 | | |
| Female | 18 (23.08) | 60 (76.92) | 78 (23.56) | 1.51 (0.84–2.73) | | | |
| **Type of Residence** | | | | | | | |
| Rural | 51 (24.29) | 159 (75.71) | 210 (63.44) | **1.91 (1.18–3.10)** | **0.009** | 0.60 (0.22-1.61) | 0.312 |
| Urban | 46 (38.02) | 75 (61.98) | 121 (36.56) | Ref | | Ref | |
| **Marital status** | | | | | | | |
| Married | 81 (28.32) | 205 (71.68) | 286 (86.40) | 1.40 (0.72–2.71) | 0.323 | | |
| Unmarried | 16 (35.56) | 29 (64.44) | 45 (13.60) | Ref | | | |
| **Education** | | | | | | | |
| Formal | 55 (26.83) | 150 (73.17) | 205 (61.93) | Ref | 0.208 | | |
| Informal | 42 (33.33) | 84 (66.67) | 126 (38.07) | 0.73 (0.45–1.19) | | | |
| **Occupation** | | | | | | | |
| Daily wage/casual | 13 (18.31) | 58 (81.69) | 71 (21.45) | **2.13 (1.11–4.10)** | **0.024** | **6.15 (1.81-20.85)** | **0.004** |
| Others | 84 (32.31) | 176 (67.69) | 260 (78.55) | **Ref** | | **Ref** | |
| **Income status (n = 319)** | | | | | | | |
| Below poverty line (BPL) | 75 (25.60) | 218 (74.40) | 293 (91.85) | **3.39 (1.50–7.66)** | **0.003** | **13.89 (3.37-57.17)** | **<0.001** |
| Above poverty line (APL) | 14 (53.85) | 12 (46.15) | 26 (8.15) | **Ref** | | **Ref** | |
| **Health Insurance Enrolment Status** | | | | | | | |
| Yes | 12 (46.15) | 14 (53.85) | 26 (7.85) | Ref | 0.054 | | |
| No | 85 (27.87) | 220 (72.13) | 305 (92.15) | 2.22 (0.98–4.99) | | | |
| **Facility first visited** | | | | | | | |
| Private | 30 (13.89) | 186 (86.11) | 216 (65.26) | **8.65 (5.07–14.77)** | **<0.001** | **17.27 (5.86-50.94)** | **<0.001** |
| Public | 67 (58.26) | 48 (41.74) | 115 (34.74) | **Ref** | | **Ref** | |
| **Distance of nearest public health facility from home (km)** | | | | | | | |
| ≤3 | 42 (31.11) | 93 (68.89) | 135 (40.79) | Ref | 0.549 | | |
| >3 | 55 (28.06) | 141 (71.94) | 196 (59.21) | 1.16 (0.72–1.87) | | | |
| **Smear status (n = 277)** | | | | | | | |
| Positive | 59 (25.11) | 176 (74.89) | 235 (84.84) | **2.46 (1.25–4.84)** | **0.009** | 0.65 (0.24-1.71) | 0.386 |
| Negative | 19 (45.24) | 23 (54.76) | 42 (15.16) | **Ref** | | Ref | |
| **No. of Times HCF visited** | | | | | | | |
| ≤3 | 76 (52.05) | 70 (47.95) | 146 (44.11) | **Ref** | **<0.001** | **Ref** | **0.015** |
| >3 | 21 (11.35) | 164 (88.65) | 185 (55.89) | **8.47 (4.85–14.82)** | | **3.00 (1.23-7.31)** | |
| **Place of TB diagnosis** | | | | | | | |
| Private | 17 (58.62) | 12 (41.38) | 29 (8.76) | **0.25 (0.11–0.55)** | **<0.001** | **0.07 (0.01-0.29)** | **<.001** |
| Public | 80 (26.49) | 222 (73.51) | 302 (91.24) | **Ref** | | **Ref** | |
| **Drug resistance** | | | | | | | |
| Yes | 4 (18.18) | 18 (81.82) | 22 (6.65) | 1.94 (0.63–5.88) | 0.243 | | |
| No | 93 (30.10) | 216 (69.90) | 309 (93.35) | Ref | | | |
| **Knowledge score towards TB symptoms** | | | | | | | |
| Poor | 61 (26.75) | 167 (73.25) | 228 (68.88) | 1.47 (0.89–2.43) | 0.13 | | |
| Good | 36 (34.95) | 67 (65.05) | 103 (31.12) | Ref | | | |

*(Continued)*

**Table 8.** (Continued)

| Variables | ≤7 (n = 97) | >7 (n = 234) | Total (n = 331) | Unadjusted OR (95% CI) | p-value | AOR (95% CI) | p-value |
|---|---|---|---|---|---|---|---|
| **Stigma Score** | | | | | | | |
| Low | 42 (33.87) | 82 (66.13) | 124 (37.46) | Ref | 0.159 | | |
| High | 55 (26.57) | 152 (73.43) | 207 (62.54) | 1.42 (0.87–2.29) | | | |
| **Smoking status** | | | | | | | |
| Yes | 45 (26.32) | 126 (73.68) | 171 (51.66) | 1.35 (0.84–2.17) | 0.217 | | |
| No | 52 (32.50) | 108 (67.50) | 160 (48.34) | Ref | | | |
| **Alcoholic status** | | | | | | | |
| Yes | 61 (29.61) | 145 (70.39) | 206 (62.24) | 0.96 (0.59–1.57) | 0.875 | | |
| No | 36 (28.80) | 89 (71.20) | 125 (37.76) | Ref | | | |

* Only variables significant in univariate analysis were included in the multivariable model; blanks indicate variables not included.

documented shorter delays ranging from 5 to 24 days [2,14,17,27,42–48], while others reported longer delays of 37–59 days [15,18,49–51].

In the multivariable model, informal education (i.e., lack of formal schooling) (AOR: 1.95; 95% CI: 1.02–3.71) and poor knowledge of TB symptoms (AOR: 31.40; 95% CI: 3.46–284.74) were independently associated with patient delay exceeding 30 days. Patients with informal education were nearly twice as likely to experience diagnostic delays compared with those who had formal education. Similar findings have been reported in studies from India [12,14,28,52], where illiteracy or lower education levels were identified as risk factors for delay, likely due to limited awareness, poor health literacy, and slower care-seeking behavior. In our study, poor knowledge of TB was also significantly associated with delayed diagnosis, consistent with previous reports [14,27,28,53–55] that demonstrated a strong link between inadequate TB knowledge and patient delay.

These findings highlight the need for sustained and innovative awareness initiatives by public health authorities to improve community understanding of TB, dispel misconceptions, and promote timely care-seeking. Misconceptions about TB curability and negative perceptions of DOTS services substantially contribute to diagnostic delays, while beliefs in supernatural causes and fear of stigma may lead patients to conceal symptoms and rely on self-medication or alternative treatments. Targeted awareness efforts are therefore essential to strengthen TB knowledge and encourage early engagement with formal health services [28]. Similar patterns have been reported in Mysuru, where Rajeswari et al. identified poor knowledge and lack of awareness of TB symptoms as major contributors to patient delay and recommended focused health education to facilitate earlier consultation. Likewise, Kumar et al. found that limited awareness, consultation with informal or traditional healers, and social stigma were key drivers of delayed care-seeking, noting that patients often approached unqualified providers first due to poverty, stigma, and restricted access to formal healthcare, further prolonging diagnosis and treatment initiation.

Although our study initially identified associations between age, poverty or income, occupation, stigma, and smoking with patient delay, none remained significant after multivariable adjustment, suggesting that these factors may operate indirectly through social or behavioral pathways. For example, socioeconomic status may influence access to health information and informal education, which in turn affects TB-related knowledge levels and symptom recognition. Likewise, smoking behavior and occupational exposure may shape personal perceptions of respiratory symptoms, potentially normalizing or dismissing early TB-related signs. Stigma may also discourage early disclosure or presentation, but its effect may be captured through psychosocial awareness and TB knowledge variables. In contrast, studies from India and other settings have consistently reported these variables as key determinants of diagnostic delay. Older age, low income, and manual occupations have been linked to longer delays due to restricted healthcare access and livelihood constraints [11,56–58], while stigma and smoking have been shown to discourage timely care-seeking and treatment adherence

**Table 9. Factors associated with total delay among tuberculosis patients (n = 331).**

| Variables | ≤37 (n = 140) | >37 (n = 191) | Total (n = 331) | Unadjusted OR (95% CI) | p-value | AOR (95% CI) | p-value |
|---|---|---|---|---|---|---|---|
| **Age group** | | | | | | | |
| 19-60 Yrs. | 114 (43.35) | 149 (56.65) | 263 (79.46) | Ref | 0.301 | | |
| >60 Yrs. | 26 (38.24) | 42 (61.76) | 68 (20.54) | 1.34 (0.77–2.32) | | | |
| **Sex** | | | | | | | |
| Male | 104 (41.11) | 149 (58.89) | 253 (76.44) | Ref | 0.431 | | |
| Female | 36 (46.15) | 42 (53.85) | 78 (23.56) | 0.81 (0.48–1.36) | | | |
| **Type of Residence** | | | | | | | |
| Rural | 76 (36.19) | 134 (63.81) | 210 (63.44) | **1.98 (1.25–3.12)** | **0.003** | 0.51 (0.20-1.28) | 0.156 |
| Urban | 64 (52.89) | 57 (47.11) | 121 (36.56) | **Ref** | | Ref | |
| **Marital status** | | | | | | | |
| Married | 104 (36.36) | 182 (63.64) | 286 (86.4) | **7.00 (3.24–15.10)** | **<0.001** | 4.34 (1.11-16.85) | **0.034** |
| Unmarried | 36 (80.00) | 9 (20.00) | 45 (13.6) | **Ref** | | **Ref** | |
| **Education** | | | | | | | |
| Formal | 104 (50.73) | 101 (49.27) | 205 (61.93) | **Ref** | **<0.001** | Ref | **0.005** |
| Informal | 36 (28.57) | 90 (71.43) | 126 (38.07) | **2.57 (1.60–4.13)** | | 4.91 (1.63-14.75) | |
| **Occupation** | | | | | | | |
| Daily wage/casual | 17 (23.94) | 54 (76.06) | 71 (21.45) | 2.85 (1.57–5.18) | <0.001 | 2.06 (0.71-6.01) | 0.182 |
| Others | 123 (47.31) | 137 (52.69) | 260 (78.55) | Ref | | Ref | |
| **Income status (n = 319)** | | | | | | | |
| Below poverty line (BPL) | 106 (36.18) | 187 (63.82) | 293 (88.52) | **21.16 (4.90–91.34)** | **<0.001** | 7.14 (0.99-51.54) | 0.051 |
| Above poverty line (APL) | 24 (92.31) | 2 (7.69) | 26 (7.85) | **Ref** | | Ref | |
| **Health Insurance Enrolment Status** | | | | | | | |
| Yes | 11 (42.31) | 15 (57.69) | 26 (7.85) | Ref | 0.999 | | |
| No | 129 (42.30) | 176 (57.70) | 305 (92.15) | 1.00 (0.44–2.25) | | | |
| **Facility first visited** | | | | | | | |
| Private | 71 | 145 | 216 | **3.06 (1.91–4.89)** | **<0.001** | **15.80 (5.45-45.78)** | **<.001** |
| Public | 69 | 46 | 115 | **Ref** | | **Ref** | |
| **Distance of nearest public health facility from home (km)** | | | | | | | |
| ≤3 | 65 (48.15) | 70 (51.85) | 135 (40.79) | Ref | 0.074 | | |
| >3 | 75 (38.27) | 121 (61.73) | 196 (59.21) | 1.50 (0.96–2.33) | | | |
| **Smear status (n = 277)** | | | | | | | |
| Positive | 78 (33.19) | 157 (66.81) | 235 (71) | 0.98 (0.15-3.13) | 0.985 | | |
| Negative | 42 (100.00) | 0 (0.00) | 42 (12.69) | Ref | | | |
| **No. of Times HCF visited** | | | | | | | |
| ≤3 | 120 | 26 | 146 | **Ref** | **<0.001** | **Ref** | **<.001** |
| >3 | 20 | 165 | 185 | **38.07 (20.30–71.39)** | | **24.80 (10.34-59.50)** | |
| **Place of TB diagnosis** | | | | | | | |
| Private | 27 (93.10) | 2 (6.90) | 29 (8.76) | **0.04 (0.01–0.19)** | **<0.001** | 0.13 (0.01-1.00) | 0.051 |
| Public | 113 (37.42) | 189 (62.58) | 302 (91.24) | **Ref** | | Ref | |
| **Drug resistance** | | | | | | | |
| Yes | 5 (22.73) | 17 (77.27) | 22 (6.65) | 2.64 (0.94–7.33) | 0.063 | | |
| No | 135 (43.69) | 174 (56.31) | 309 (93.35) | Ref | | | |
| **Knowledge score towards TB symptoms** | | | | | | | |
| Poor | 65 (28.51) | 163 (71.49) | 228 (68.88) | **6.71 (3.99–11.30)** | **<0.001** | 3.74 (0.71-19.68) | 0.119 |
| Good | 75 (72.82) | 28 (27.18) | 103 (31.12) | **Ref** | | Ref | |

*(Continued)*

**Table 9.** (Continued)

| Variables | ≤37 (n = 140) | >37 (n = 191) | Total (n = 331) | Unadjusted OR (95% CI) | p-value | AOR (95% CI) | p-value |
|---|---|---|---|---|---|---|---|
| **Stigma Score** | | | | | | | |
| Low | 81 (65.32) | 43 (34.68) | 124 (37.46) | **Ref** | **<0.001** | Ref | 0.514 |
| High | 59 (28.50) | 148 (71.50) | 207 (62.54) | **4.72 (2.93–7.61)** | | 0.57 (0.71-19.68) | |
| **Smoking status** | | | | | | | |
| Yes | 55 (32.16) | 116 (67.84) | 171 (51.66) | **2.39 (1.52–3.74)** | **<0.001** | 1.06 (0.10-3.04) | 0.935 |
| No | 85 (53.13) | 75 (46.88) | 160 (48.34) | **Ref** | | Ref | |
| **Alcoholic status** | | | | | | | |
| Yes | 75 (36.41) | 131 (63.59) | 206 (62.24) | **1.89 (1.20–2.97)** | **0.006** | 1.50 (0.25-4.51) | 0.59 |
| No | 65 (52.00) | 60 (48.00) | 125 (37.76) | **Ref** | | Ref | |

* Only variables significant in univariate analysis were included in the multivariable model; blanks indicate variables not included.

[59–60]. These findings underscore the importance of locally tailored interventions that address interconnected socioeconomic and behavioral barriers to early tuberculosis diagnosis.

In India, where poverty remains a major challenge, individuals with tuberculosis (TB) face multiple socioeconomic barriers to diagnosis and treatment, including food insecurity, high travel costs, and income loss during illness [61–63]. As with leprosy and HIV, TB diagnosis and the public use of anti-TB medication can lead to social exclusion, perceived stigma, and concealment of the disease, contributing to delays in treatment initiation [64–67]. Mistreatment by family members and a prevailing culture of blame and shame, particularly within certain social and caste groups, further undermine treatment adherence and facilitate continued transmission [28,68]. In our study, 68% of participants had low awareness of TB symptoms, and 62.5% reported stigma or fear of isolation as reasons for delay, underscoring the need for sustained awareness activities in this region. Similar patterns of limited awareness and fear-driven delays have been reported in India [14,27,69], Ethiopia [31,54,70,71], and other settings [72,73]. Interventions such as peer-support programmes to address TB-related stigma, together with enhanced collaboration between non-governmental organizations and government health services, may help improve treatment outcomes in India [74].

## Factors associated with health system delay

In our study, the median health system delay was 12 days. This finding aligns with several studies [15,26–28,44,46], which reported a median delay of approximately 12 (±5) days. In contrast, some studies have documented longer system delays ranging from 18 to 56 days [2,12,13,17,18,36,42,43,45,50,51], whereas others have reported much shorter delays of 2–3 days [14,39].

In our multivariable model, being a daily wage or casual worker (AOR: 6.15; 95% CI: 1.81–20.85) was significantly associated with health system delay exceeding 7 days. Similar findings were reported by Van Ness et al. in South India, where daily wage earners faced longer delays due to income loss and limited opportunities for healthcare access [75,76], and by Mesfin et al. in Ethiopia, who also noted prolonged delays among patients engaged in informal or daily wage work [77]. Participants belonging to the below-poverty-line (BPL) category (AOR: 13.89; 95% CI: 3.38–57.17) were more likely to experience longer delays, consistent with studies from Karnataka, India [42,78,79], Ethiopia [80], and elsewhere [59,81,82], which highlighted low socioeconomic status as a key barrier to timely diagnosis and healthcare access. Although daily wage occupation and below-poverty-line status are patient-level characteristics, their association with health system delay reflects how socioeconomic vulnerability influences patients' ability to navigate healthcare services after initial care-seeking. Income insecurity and financial constraints may limit repeated visits, access to diagnostics,

transportation, and referrals. These findings therefore highlight inequities in patients' experiences of health system processes rather than failures of the health system itself.

Patients who first sought care at private healthcare facilities (AOR: 17.28; 95% CI: 5.86–50.94) had significantly higher odds of delay, reflecting findings from India and other settings where initial visits to private practitioners often resulted in misdiagnosis and delayed referral to government TB services [13,26–28,38,42,83–86]. Those who visited healthcare facilities more than three times before diagnosis (AOR: 3.01; 95% CI: 1.23–7.32) also experienced greater health system delays, in line with studies from India, Nepal, Ethiopia, and other regions [12,13,18,26–28,56,87,88]. Interestingly, being diagnosed at a private facility was protective (AOR: 0.072; 95% CI: 0.018–0.30), consistent with Arinaminpathy et al. [89], who observed faster treatment initiation in private healthcare settings, likely due to shorter administrative procedures compared with public facilities [90]. This seemingly contradictory pattern suggests different processes operating at two stages of care-seeking. Initial consultations at private clinics or non-specialist clinics can contribute to diagnostic delays due to low clinical suspicion of TB, symptomatic rather than investigative management, and limited access to microbiological testing. Patients may receive multiple rounds of nonspecific antibiotics or cough suppressants before being referred for TB evaluation, prolonging the diagnostic interval. However, once TB is formally diagnosed within a private setting, treatment initiation may occur rapidly due to streamlined administrative procedures, and faster referral or linkage to treatment services. Thus, private sector involvement may hinder early recognition of TB but facilitate rapid initiation of treatment once diagnosis is confirmed.

In India, private healthcare providers are often more accessible and trusted by patients than public facilities [90–92]. However, many private providers do not offer TB diagnostic services. To ensure timely identification of all undiagnosed TB cases, it is therefore essential to integrate private sector providers into the national health system.

## Factors associated with total delay

In our study, the median total delay from symptom onset to treatment initiation was 43 days, consistent with previous reports [17,43,46], which documented a median delay of approximately 43 (±5) days. In contrast, some studies have reported shorter delays of 19–36 days [12,14,39,44,47], while others reported longer delays of 49–104 days [15,18,26,36,42,49]. These variations underscore the role of local healthcare access, patient awareness, and health system efficiency in timely TB diagnosis and treatment.

In the adjusted analysis, several factors were independently associated with prolonged total delay. Patients with informal education (AOR: 4.91; 95% CI: 1.64–14.76) were more likely to experience delays in seeking TB care, consistent with previous studies showing that lack of formal education is a significant risk factor for delayed care-seeking, likely due to lower health literacy and awareness [28,93]. Patients who initially consulted a private healthcare facility and those who visited more than three healthcare providers before receiving a TB diagnosis were also significantly more likely to experience prolonged delays (AOR: 15.80; 95% CI: 5.45–45.78 and AOR: 24.81; 95% CI: 10.34–59.50, respectively), in line with previous reports [13,26–28,94]. Interestingly, being married (AOR: 4.34; 95% CI: 1.12–16.85) was associated with longer delays, possibly because married individuals often have greater family responsibilities, limiting the time available for seeking care and leading them to prioritize household needs over their own health.

Stigma remains a complex and persistent barrier to timely TB diagnosis, often contributing to delayed disclosure and care-seeking. In our study, however, high perceived stigma did not remain significant in the adjusted patient-delay model; although an association was observed in unadjusted analyses, the effect was attenuated after accounting for knowledge and education, suggesting that stigma may influence delays indirectly through these factors. Similar observations were reported by Shah et al. in Gujarat, India [95], where patients struggled to accept a TB diagnosis and believed that affected individuals must live and eat separately, highlighting the need for counseling on diet, lifestyle, and adherence. Stigma should therefore not be overlooked in programmatic interventions; reducing it through integrated community education remains essential, as shown in India and comparable settings [14,27,95–98]. Additionally, integrating non-NTP providers

into the national TB programme—through direct service involvement or referral training—has been shown to improve TB detection, notification, and timely treatment, including for drug-resistant TB, in similar Indian contexts.

The body of research on tuberculosis diagnostic delay in India shows substantial variability but consistently indicates prolonged median total delays, generally ranging from about 19 to over 120 days across regions and healthcare settings. Many studies report delays exceeding one to two months and often surpassing the critical 37-day threshold, demonstrating that a large proportion of patients experience considerably delayed diagnosis [11,12,14,17,26,27,36,42–44,47,83,85,99–110]. This pattern reflects persistent barriers to early detection, including limitations in healthcare access, patient-level delays, and diagnostic inefficiencies. More recent studies show somewhat shorter delays, with some reporting median values under 40 days, suggesting incremental improvements in awareness and diagnostic pathways; however, delays remain a significant public health concern. Comparisons with other low- and middle-income countries place India within a similar range but reinforce the need for comprehensive strategies to reduce diagnostic delay and improve timely treatment initiation [18,38,39,46,48,49,56,70,81,88,111–127]. Overall, the accumulated evidence highlights diagnostic delay as a continuing challenge and priority target for strengthening TB control in India.

## Strengths and limitations of the study

This study demonstrates several key strengths that enhance its contribution to tuberculosis research. It includes a diverse sample that enhances the reliability and internal consistency of the findings in our regional setting. The use of validated, culturally adapted questionnaires based on World Health Organization methodologies ensures data quality and comparability. Rigorous multivariable statistical analyses enable identification of independent predictors of diagnostic and treatment delays, providing nuanced insights into complex barriers. Clear operational definitions grounded in established clinical thresholds add methodological rigor and improve reproducibility. Furthermore, comprehensive quality assurance measures, including expert review and cross-verification with patient records, underpin the robustness of the dataset. The integration of psychosocial factors such as stigma and TB knowledge alongside clinical and sociodemographic data offers a holistic perspective critical for informing effective, targeted public health interventions in high-burden settings. Cases were confirmed using the health management information system, and the questionnaire was adapted from the WHO Multi-Country Study on TB Diagnosis and Treatment Delays which make the data comparable to other studies using this validated tool.

While this study offers valuable insights, certain limitations should be acknowledged to guide interpretation and inform future research. The single-center, facility-based design limits external validity and generalizability beyond the Mysuru context, and the cross-sectional nature precludes causal inference between risk factors and delays in diagnosis or treatment. Reliance on patient recall of symptom onset and health-seeking behavior may introduce recall bias, affecting delay estimates. The focus on patient-side barriers means system- and provider-level contributors—such as diagnostic capacity or referral patterns—may not be fully captured, and the quantitative assessment of stigma does not explore how stigma shapes health-seeking behavior. Unmeasured confounders, including mental health, comorbidities, or variations in healthcare delivery, were not comprehensively addressed. Although recommendations are proposed, their practical impact, feasibility, and cost-effectiveness were not directly evaluated. Some multivariable model estimates yielded very large AORs with wide 95% CIs, likely reflecting sparse data or quasi-complete separation, limiting precision; future studies with larger datasets or penalized methods (e.g., Firth regression) may provide more stable estimates. Continuous variables, such as knowledge and stigma scores, were dichotomized at the median, potentially causing information loss and reduced power; future work using continuous modeling or multiple categories (e.g., tertiles, quartiles) may improve estimation. Finally, using $p \leq 0.05$ for including variables from bivariate to multivariable models may have excluded relevant confounders, reducing model stability and risking omitted-variable bias; future analyses could benefit from more inclusive criteria (e.g., $p \leq 0.20$) and prioritizing variables based on theoretical or clinical relevance rather than statistical screening alone.

## Implications

This study highlights several actionable interventions to reduce TB diagnosis and treatment delays. Awareness campaigns targeting low education and poor knowledge, including community outreach, mass media, and peer educators, can promote early care-seeking. Training and integrating private and informal providers into the National TB Elimination Program, with emphasis on early referral and use of diagnostics, can shorten system delays. Strengthening referral networks, improving peripheral diagnostic capacity, and monitoring the number of visits before diagnosis can streamline care pathways. Support for economically vulnerable patients, such as travel assistance, food incentives, and expanded insurance coverage, can reduce financial barriers. Although stigma was not an independent predictor, high perceived stigma suggests that counseling, community sensitization, and destigmatizing messages remain important. Routine monitoring of patient, health system, and total delays by provider type, visit frequency, and socioeconomic factors can guide targeted interventions and improve TB program performance.

## Conclusion

The study revealed that nearly two-thirds of the participants experienced delays in diagnosis of tuberculosis in Mysuru, South India. These delays were primarily attributed to poor TB-related knowledge, high perceived stigma, informal education, poverty, initial consultation with private or informal healthcare providers, and visiting multiple healthcare facilities after symptom onset. Patient delay was predominant, with a median of 30 days, while the health system delay was 12 days, resulting in a total median delay of 43 days. This indicates that both patient- and health system-related factors contributed substantially to diagnostic and treatment delays. Reducing TB diagnostic and treatment delays in this setting requires multi-pronged interventions: strengthening community awareness, integrating private providers within the National TB Elimination Programme, enhancing peripheral diagnostic capacity, and providing social and economic support to vulnerable populations. Future studies should explore health-system bottlenecks from provider perspectives and employ prospective designs to minimize recall bias. To achieve India's TB elimination goals, interventions must extend beyond biomedical strategies toward equitable, patient-centered, and timely TB care delivery through improved coordination and sustained public–private collaboration.

## Supporting information

**S1 File. Semi-structured questionnaire used for data collection.**
(PDF)

## Acknowledgments

The authors express their sincere gratitude to the DTO, Mysuru, and the STO, Karnataka, for granting permission to conduct this study at the PKTB Sanatorium, Mysuru. We also extend our appreciation to the medical officers, laboratory staff, and health visitors for their support during data collection. Our heartfelt thanks to all the patients who participated and shared their experiences.

## Author contributions

**Conceptualization:** Veerabhadra Swamy G S, Mahesh Padukudru Anand, Mahadevaiah Neelambike Sumana, N B Ramachandra, Prashanth Chikkahonnaiah.

**Data curation:** Veerabhadra Swamy G S, Chinchana Shylaja Eshwarappa, Yogeesh D Maheshwarappa.

**Formal analysis:** Veerabhadra Swamy G S, Mahadevaiah Neelambike Sumana, N B Ramachandra, Prashanth Chikkahonnaiah, Jyothi M N.

**Investigation:** Chinchana Shylaja Eshwarappa, Yogeesh D Maheshwarappa.

**Methodology:** Veerabhadra Swamy G S, Mahesh Padukudru Anand, Mahadevaiah Neelambike Sumana.

**Resources:** Prashanth Chikkahonnaiah, Jayaraj B S.

**Software:** Veerabhadra Swamy G S, Jyothi M N.

**Supervision:** Mahesh Padukudru Anand, Mahadevaiah Neelambike Sumana.

**Validation:** Mahesh Padukudru Anand, Jyothi M N.

**Visualization:** Veerabhadra Swamy G S, Chinchana Shylaja Eshwarappa, Yogeesh D Maheshwarappa.

**Writing – original draft:** Veerabhadra Swamy G S.

**Writing – review & editing:** Veerabhadra Swamy G S, Mahesh Padukudru Anand, Mahadevaiah Neelambike Sumana, Ranjitha Shankaregowda.

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
