## [Decision Letter · Decision Letter 0]

24 Nov 2025

Dear Dr. Sumana,

Thank you for submitting your manuscript to PLOS ONE. After careful consideration, we feel that it has merit but does not fully meet PLOS ONE’s publication criteria as it currently stands. Therefore, we invite you to submit a revised version of the manuscript that addresses the points raised during the review process.

We look forward to receiving your revised manuscript.

Kind regards,

Yasir Alvi

Academic Editor

PLOS ONE

Reviewers' comments:

Reviewer's Responses to Questions

**Comments to the Author**

1. Is the manuscript technically sound, and do the data support the conclusions?

Reviewer #1: No

Reviewer #2: Partly

2. Has the statistical analysis been performed appropriately and rigorously?

Reviewer #1: No

Reviewer #2: Yes

3. Have the authors made all data underlying the findings in their manuscript fully available?

Reviewer #1: Yes

Reviewer #2: Yes

4. Is the manuscript presented in an intelligible fashion and written in standard English?

Reviewer #1: No

Reviewer #2: Yes

Reviewer #1: The manuscript presents a timely and important study on the factors affecting diagnostic and treatment delays in tuberculosis in Mysuru, South India. To further strengthen the manuscript and enhance the clarity and robustness of its conclusions, we recommend incorporating the following considerations and revisions.

Methodology

•The manuscript presents a contradiction by describing the study as "community-based" in the abstract and methods section, while simultaneously acknowledging it as "single-center, facility-based" in the limitations. Recruitment exclusively from a tertiary-care hospital inherently defines the study as facility-based, which impacts the generalizability of the results. This terminology should be clarified for consistency to accurately reflect the source of participants and the implications for external validity.

•While "Diagnostic delay" and "Treatment delay" are adequately defined, the relationship between these and the "Patient delay" and "Health system delay" components, which sum to "Total delay," needs explicit clarification in the text. Relying solely on Figure 2 for this crucial relationship could lead to misinterpretation. The definitions should clearly state how "Diagnostic delay" and "Treatment delay" are situated within or relate to "Patient delay" and "Health system delay," respectively.

Results and Statistical Analysis

•The multivariable logistic regression models frequently yield extremely large Adjusted Odds Ratios (AORs) accompanied by exceptionally wide 95% Confidence Intervals (CIs). For instance, an AOR of 31.40 (95% CI: 3.46–284.74) for poor TB knowledge or 24.81 (95% CI: 10.34–59.50) for visiting more than three facilities indicates significant model instability, likely due to small cell sizes (quasi-complete separation). Presenting such point estimates without acknowledging this high imprecision is misleading. The discussion notably omits the interpretation of this instability. Future analyses may consider more robust statistical methods, such as Firth regression or Exact Logistic Regression to produce more stable and reliable estimates.

•The practice of dichotomising continuous variables, such as knowledge and stigma scores, at their median is statistically suboptimal. This approach results in the loss of valuable data. Analysing these variables as continuous predictors (with appropriate checks or transformations) or utilising multiple ordered categories (e.g., quartiles) would provide a more complete and statistically sound understanding of their relationships with the outcomes.

•The use of a p-value-based stepwise selection approach (p≤0.05 in bivariate analysis for inclusion in multivariable models) can lead to model instability, omitted-variable bias, and potentially spurious findings. A more robust and recommended approach involves selecting variables based on theoretical relevance or employing a less stringent univariate screening threshold (e.g., p≤0.20) to ensure the inclusion of important confounders in the multivariable models.

Discussion

•The discussion notes that factors like age, poverty, occupation, stigma, and smoking, initially associated with patient delay, lost significance after multivariable adjustment, suggesting indirect effects. To strengthen this observation, the discussion should elaborate on how these factors might operate indirectly or identify the most likely mediators in the study's specific context. For example, exploring whether they primarily influence informal education or TB-related knowledge could provide deeper contextual insights.

•The discussion presents an intriguing paradox: while initially seeking care at private facilities is associated with significantly higher odds of delay (AOR: 17.28), being diagnosed at a private facility appears to be protective against delay in treatment initiation (AOR: 0.072). The discussion should provide a more explicit and detailed explanation of this distinction. It needs to clarify whether this implies that initial private consultations often involve misdiagnosis or a lack of TB services, thus causing delays. However, once a diagnosis is established within a private setting, the administrative process for treatment initiation is faster.

Reviewer #2: Comment

Clarification is required as the study was done at at the PKTB) Sanatorium, not in the community, hence study design is center based not the community based.

2. Abstract/Result author has mentioned system delay was linked to daily wage occupation (AOR: 6.15; 95% CI: 1.81–20.85

Comment: What were the attributes or factors considered under health system delay , as occupation and below poverty-line status, marital status etc cannot be considered as a risk factor for health system delay as it is more related to delays at patient level.

3. Abstract /conclusion, author has mentioned Nearly two-thirds of TB patients experienced diagnostic or treatment delay

Comment: Need to explain whether 2/3rd of TB patients experienced diagnostic or treatment delay, as both are different things, factors responsible for delay and putting on treatment are different.

4. MATERIALS AND METHODS , author has mentioned, Study designed mentioned is retrospective and the study population comprised newly diagnosed TB patients aged ≥18 years who gave written informed consent and were available for interviews during the study period.

Comment: Why author has mentioned retrospective study if new cases were enrolled.. selection criteria needs clarification in terms of inclusion/ exclusion criteria and sampling technique used, as the Centre is offering IPD/OPD based services From where the cases were enrolled.

5. MATERIALS AND METHODS /Sample size and selection, author has mentioned prevalence of approximately 40.5%,a margin of error of 5%

Comment: This formula fits more for community based study not the Centre based one, current study is Centre based. This margin error was relative precision or absolute precision, needs clarification.

6. Results, author has given description for Sociodemographic characteristics of the study participants.

Comment: Table 1 can be removed to avoid duplication

Clinical characteristics/

7. In the result , Factors associated with patient/diagnostic/treatment/health system delay has been reflected.

Comment : 1)Needs to figure out of total 331 interviewed cases, what were presenting symptoms and the risk factors for delays ( patients/health system) in context for microbiologically confirmed cases of pulmonary and extra pulmonary Tb cases, and factors should analyzed separately, for all types of delays patient/diagnostic/treatment /health system delays for pulmonary and extra pulmonary Tb cases

8. Comment: Table 1 to 5 can be omitted to avoid duplication as summarized through text only, table number 8 where Factors associated with health system delay among tuberculosis patient has been depicted.

9. Table/s can be placed at the end after the references .

10. Discussion is i is lengthy, needs to make concise and crispy

11. Strengths and limitations of the study : It includes a large, diverse sample that strengthens the reliability and generalizability of findings within the regional context.

Comment : This is single center based Study with sample size of 331 so results can be generalized so lacks external validity.

**Do you want your identity to be public for this peer review?** For information about this choice, including consent withdrawal, please see our Privacy Policy

Reviewer #1: **Yes:** Prof. (Dr) Arun Kokane

Reviewer #2: **Yes:** Dr. Rashmi Sharma

---

## [Author Response · Author response to Decision Letter 1]

14 Dec 2025

We sincerely thank the Editor and Reviewers for their constructive and valuable comments. We have carefully addressed each point in the revised manuscript. Detailed responses to every comment are provided below, and corresponding changes have been incorporated into the text. We believe these revisions have strengthened the clarity, methodology, and overall quality of the manuscript. We respectfully request the Editor and Reviewers to consider the revised version for further evaluation. We also remain fully open to making any additional revisions recommended by the Editor or Reviewers. We greatly value their guidance and are willing to incorporate any further changes required to meet the journal’s standards and policies.

---

## [Decision Letter · Decision Letter 1]

2 Jan 2026

Dear Dr. Sumana,

Thank you for submitting your manuscript to PLOS ONE. After careful consideration, we feel that it has merit but does not fully meet PLOS ONE’s publication criteria as it currently stands. Therefore, we invite you to submit a revised version of the manuscript that addresses the points raised during the review process.

We look forward to receiving your revised manuscript.

Kind regards,

Yasir Alvi

Academic Editor

PLOS One

Journal Requirements:

Reviewers' comments:

Reviewer's Responses to Questions

**Comments to the Author**

Reviewer #1: All comments have been addressed

Reviewer #2: All comments have been addressed

2. Is the manuscript technically sound, and do the data support the conclusions?

Reviewer #1: Yes

Reviewer #2: Yes

3. Has the statistical analysis been performed appropriately and rigorously?

Reviewer #1: Yes

Reviewer #2: Yes

4. Have the authors made all data underlying the findings in their manuscript fully available?

Reviewer #1: Yes

Reviewer #2: Yes

5. Is the manuscript presented in an intelligible fashion and written in standard English?

Reviewer #1: Yes

Reviewer #2: Yes

Reviewer #1: The manuscript presents a timely and important study on the factors affecting diagnostic and treatment delays in tuberculosis in Mysuru, South India. I recommend the publication of this manuscript.

Reviewer #2: Following are the revised comments for Factors Affecting Diagnostic and Treatment Delays in Tuberculosis: A Cross-Sectional Study in Mysuru, South India

1. Title reflects health system delays however patient related delay is missing. for health system delays which mainly includes diagnosis and treatment delays or slowing down timely diagnosis and treatment after a patient seeks care. Which is within healthcare infrastructure, inadequate resources, provider practices, and patient flow, including limited diagnostic tools (like X-ray, smear tests), poor referral systems between public/private sectors, insufficient clinician awareness of diseases (like TB), geographical barriers, long wait times, author has mentioned some of them correctly that Health system delay was linked to first seeking care at private facilities (AOR: 17.28; 95% CI: 5.86–50.94), and visiting more than three healthcare facilities before diagnosis, ) however as per earlier comment that Health system delay was linked to daily wage occupation (AOR: 6.15; 95% CI: 1.81–20.85), below-poverty-line economic status (AOR: 13.89; 95% CI: 3.38–57.17) is still leading to confusion to consider it as patient related delay rather than Health system delay.

2. In results about Sociodemographic characteristics of the study participants have been already mentioned as per earlier comments to avoid repetition table 1 can be removed.

3. Regarding Clinical characteristics in result section as per earlier remark author has mentioned only symptoms related to pulmonary TB, however there were 14 Extra pulmonary TB cases it would be interesting to know what was the presenting or clinical finding.

**Do you want your identity to be public for this peer review?** For information about this choice, including consent withdrawal, please see our Privacy Policy

Reviewer #1: **Yes:** Prof. (Dr) Arun Kokane, AIIMS Bhopal

Reviewer #2: No

---

## [Author Response · Author response to Decision Letter 2]

12 Jan 2026

We sincerely thank the Editor and all reviewers for their careful evaluation of our manuscript and for the constructive and insightful comments. We greatly appreciate the time and effort invested in reviewing our work. The suggestions provided have helped us improve the clarity, rigor, and overall quality of the manuscript. We have addressed all comments point by point and revised the manuscript accordingly. We believe that these revisions have strengthened the manuscript and hope it is now suitable for publication.

---

## [Decision Letter · Decision Letter 2]

1 Feb 2026

Patient and health system related factors affecting diagnostic and treatment delays in tuberculosis: A cross-sectional study in Mysuru, South India

PONE-D-25-57862R2

Dear Dr. Sumana,

We’re pleased to inform you that your manuscript has been judged scientifically suitable for publication and will be formally accepted for publication once it meets all outstanding technical requirements.

Kind regards,

Yasir Alvi

Academic Editor

PLOS One

Reviewers' comments:

Reviewer's Responses to Questions

**Comments to the Author**

Reviewer #2: All comments have been addressed

2. Is the manuscript technically sound, and do the data support the conclusions?

Reviewer #2: Yes

3. Has the statistical analysis been performed appropriately and rigorously?

Reviewer #2: Yes

4. Have the authors made all data underlying the findings in their manuscript fully available?

Reviewer #2: Yes

5. Is the manuscript presented in an intelligible fashion and written in standard English?

Reviewer #2: Yes

Reviewer #2: This revised mansucript can be accepted for publications, it would be better if table one which describes sociodemographic information which is already reflected in subsequent tables and text can be ommitted to avoid duplication of information

**Do you want your identity to be public for this peer review?** For information about this choice, including consent withdrawal, please see our Privacy Policy

Reviewer #2: **Yes:** Rashmi Sharma

---

## [Editor Report · Acceptance letter]

PONE-D-25-57862R2

PLOS One

Dear Dr. Sumana,

I'm pleased to inform you that your manuscript has been deemed suitable for publication in PLOS One. Congratulations! Your manuscript is now being handed over to our production team.

Kind regards,

on behalf of

Dr. Yasir Alvi

Academic Editor

PLOS One